# Design and Testing of Mobile Laboratory for Mitigation of Gaseous Emissions from Livestock Agriculture with Photocatalysis

**DOI:** 10.3390/ijerph18041523

**Published:** 2021-02-05

**Authors:** Myeongseong Lee, Jacek A. Koziel, Wyatt Murphy, William S. Jenks, Blake Fonken, Ryan Storjohann, Baitong Chen, Peiyang Li, Chumki Banik, Landon Wahe, Heekwon Ahn

**Affiliations:** 1Department of Agricultural and Biosystems Engineering, Iowa State University, Ames, IA 50011, USA; leefame@iastate.edu (M.L.); wyatt.murphy@jetinc.net (W.M.); bjfonken@iastate.edu (B.F.); rls1@iastate.edu (R.S.); baitongc@iastate.edu (B.C.); peiyangl@iastate.edu (P.L.); cbanik@iastate.edu (C.B.); lwahe@iastate.edu (L.W.); 2Department of Chemistry, Iowa State University, Ames, IA 50011, USA; wsjenks@iastate.edu; 3Department of Animal Biosystems Sciences, Chungnam National University, Daejeon 34134, Korea; hkahn@cnu.ac.kr

**Keywords:** air pollution control, air quality, volatile organic compounds, odor, environmental technology, advanced oxidation, UV-A, titanium dioxide

## Abstract

Livestock production systems generate nuisance odor and gaseous emissions affecting local communities and regional air quality. There are also concerns about the occupational health and safety of farmworkers. Proven mitigation technologies that are consistent with the socio-economic challenges of animal farming are needed. We have been scaling up the photocatalytic treatment of emissions from lab-scale, aiming at farm-scale readiness. In this paper, we present the design, testing, and commissioning of a mobile laboratory for on-farm research and demonstration of performance in simulated farm conditions before testing to the farm. The mobile lab is capable of treating up to 1.2 m^3^/s of air with titanium dioxide, TiO_2_-based photocatalysis, and adjustable UV-A dose based on LED lamps. We summarize the main technical requirements, constraints, approach, and performance metrics for a mobile laboratory, such as the effectiveness (measured as the percent reduction) and cost of photocatalytic treatment of air. The commissioning of all systems with standard gases resulted in ~9% and 34% reduction of ammonia (NH_3_) and butan-1-ol, respectively. We demonstrated the percent reduction of standard gases increased with increased light intensity and treatment time. These results show that the mobile laboratory was ready for on-farm deployment and evaluating the effectiveness of UV treatment.

## 1. Introduction

Over the past few decades, livestock and poultry farmers have adopted new technology and have scaled up farming operations to meet society’s demand for high-quality meats, milk, eggs, and by-products. Large confined animal feeding operations (CAFOs) are common in many parts of the world. This has generated profits and jobs, but the environmental problems associated with local air quality have been exacerbated. These unwanted side-effects of animal production require sustainable solutions for the benefit of workers, rural communities, and the industry.

The U.S. National Air Emissions Monitoring Study (NAEMS) developed an accurate baseline emission database for CAFO regulation by the US EPA through the notification provisions of the Emergency Planning and Community Right-to-Know Act (EPCRA) and the Clean Air Act (CAA) [1,2]. NAEMS and the companion projects focused on monitoring emissions of odor, volatile odorous compounds (VOCs), ammonia (NH_3_), hydrogen sulfide (H_2_S), carbon dioxide (CO_2_), methane (CH_4_), the total suspended particulates (TSP), PM10, and PM2.5 in the egg, broiler, dairy, and swine production industries [1,2,3,4,5,6,7]. While the NAEMS can be used as a standard and a source of pollutants emitted from farms, there is still a need to develop and test mitigation technologies that are consistent with the socio-economic reality of CAFOs. Mitigation technologies for gaseous emissions from livestock operations could be classified amongst approximately 12 approaches, including facility barriers, biofilters, chimneys, diet manipulation, electrostatic precipitation, landscaping, oil sprinkling, pit ventilation, scrubbers, siting, urine (or feces) segregation, and UV light [8]. The overview of each mitigation technology and citations in research papers is summarized elsewhere [8,9].

Farm-scale performance data are a prerequisite for the adoption of proposed new technology. Farmers need proven technologies before agreeing on farm-scale trials. Well-intentioned, laboratory-scale experimentation cannot fully duplicate the on-farm variability. Maurer et al. (2016) summarized the current state of adoption of technologies for mitigation of gaseous emissions from livestock agriculture [9]. Only ~25% of mitigation technologies developed and tested in lab-scale have been tested in real-farm conditions. We have been scaling up the photocatalytic treatment of emissions from the lab- to pilot-scales, aiming at farm-scale readiness [10,11,12,13,14,15,16,17]. Several other research teams have also been testing UV photocatalytic technology [18,19,20,21] for the mitigation of gaseous emissions from livestock operations.

UV light treatment is a promising technology for mitigating gaseous pollutants. The use of either shorter UV wavelengths or a photocatalyst improves the mitigation effects [12,14,15]. In addition, catalytic coating type, coating dose, UV dose, relative humidity, temperature, and dust accumulation (on photocatalyst) are important variables to consider and optimize for improved reduction of targeted odorous gases [12,17]. The photocatalytic treatment has been found to show a significant reduction in odorous VOCs even after short effective treatment times that are consistent with fast-moving ventilation air on farms [10,15]. Previous studies have reported the varying effect of reducing NH_3_, H_2_S, greenhouse gases, VOCs, odor, and particulate matter (PM) with UV in livestock farm conditions [9,10,11,12,13,14,15,16,17,18,19,20].

Only a selected few studies reported on testing UV technology on a pilot scale [11,13,15] or farm-scale [18,19]. For that reason, there is a lack of information on UV doses and cost to reduce odorous gases in farm-scale conditions. In addition, depending on the wavelengths of UV light, direct exposure to the light or its by-products (e.g., ozone) generated by shorter wavelength UV (e.g., 254 nm) can be risky to workers and livestock. Our previous research showed that the intrinsically safer UV-A (365 nm) could be effective in treating NH_3_, N_2_O, ozone, selected VOCs, and odor on lab- and pilot-scales [11,12,13].

Therefore, we hypothesize that the UV-A based photocatalysis can be effective in reducing selected gaseous emissions at a much larger scale. A UV-A mobile lab is a research tool that could be used to perform on-site trials at different farms and industrial emissions sources to demonstrate UV-A performance at realistic conditions. The farmers and industry appreciate these types of trials that do not disrupt current operations while providing necessary decision-making data. This was the motivation behind the design of a self-contained mobile laboratory that can directly sample the gases from a livestock farm and carry out the evaluation of photocatalysis UV treatment and cost prior to the next logical step, i.e., scaling up and installation of UV treatment on a farm or other emissions source.

The objective of this research was to design and test a mobile laboratory for the mitigation of gaseous emissions from livestock barns with UV-A photocatalysis. To our knowledge, this is the first study to evaluate the effect of UV-A photocatalysis treatment under conditions similar to a livestock farm using a mobile laboratory of this type. We summarize the main technical requirements, constraints, approach, and performance metrics for the mobile laboratory, such as the effectiveness (measured as the percent reduction) and cost of photocatalytic treatment of air. We provide the mitigation effect for two representative odorous gases (NH_3_ and butan-1-ol) with the mobile laboratory. In addition, preliminary economic analysis for the cost of gaseous emissions treatment with LED UV-A lights was provided.

## 2. Materials and Methods

### 2.1. Requirements for Testing UV Photocatalysis at the Mobile Laboratory

The mobile laboratory (7.2 m × 2.4 m × 2.4 m exterior dimensions) was designed to evaluate the effectiveness of UV photocatalysis by directly connecting to the exhaust gases emitted from the farm (Figure 1).

The technical requirements and constraints for the mobile laboratory are summarized in Table 1. It explains the approach, the performance metric, and the location of the detailed description in the manuscript that addresses each of the five main requirements and constraints. In summary, we have implemented: (1) construction of treatment chambers capable of irradiating UV light and collecting real-time gas samples, (2) control of the UV dose, (3) control of the airflow, (4) control of the photocatalyst dose, and (5) control of airborne particulate matter.

### 2.2. Light Intensity Measurement

Light intensity was measured (Figure A3) with an ILT-1700 radiometer (International Light Technologies, Peabody, MA, USA) equipped with an NS365 filter and SED033 detector (International Light Technologies, Peabody, MA, USA). Prior to use, the radiometer and sensor were sent to the manufacturer company (International Light Technologies, Peabody, MA, USA) for factory calibration. For economic analysis, the electric power consumption was measured using a wattage meter (P3, Lexington, NY, USA).

### 2.3. Measurement of Standard Gases Concentration (NH_3_ and Butan-1-ol)

Two odorous gases were used for testing and commissioning. The butan-1-ol (a representative standard gas for VOCs and a mild odorant) and NH_3_ concentrations were measured in order to evaluate the percent reduction by UV photocatalysis treatment (Figure 2). The calibrations for both standard gases were at R^2^ > 0.99.

For NH_3_, standard gas and dry air were adjusted using a mass flow controller (FMA5400A/5500A Series, OMEGA, Norwalk, CT, USA) to make five diluted gas samples generally within the range of the target gas to be measured. In the case of NH_3_, diluted samples were collected in a Tedlar bag, and the concentration was measured using the gas monitoring system (OMS-300, Smart Control & Sensing Inc., Daejeon, Korea) equipped with electrochemical gas sensors of Membrapor Co. (Wallisellen, Switzerland). The calibration curve is drawn using the obtained voltage from the sensor and the known concentration of the diluted sample (Figure 3).

Air samples for butan-1-ol measurements were collected using 1 L glass gas sampling bulbs (Supelco, Bellefonte, PA, USA). Air samples were taken using a portable vacuum sampling pump (Leland Legacy; SKC Inc., Eighty-Four, PA, USA) with a set flow rate of 5 L/min for 3 min. Chemical analyses were completed using a solid-phase microextraction (SPME) (50/30 µm DVB/CAR/PDMS; 2 cm-long fibers, Supelco, Bellefonte, PA, USA) using static extraction for 1 h at room temperature and gas chromatography-mass spectrometry (GC-MS) system for analyses (Agilent 6890 GC; Microanalytics, Round Rock, TX, USA). The calibration for butan-1-ol is shown in Figure 4.

### 2.4. Photocatalyst (TiO_2_) Coating

TiO_2_ coating was applied in the same way as in the previous study [11]. TiO_2_ coating on the pre-cut panels for the UV reactor was carried out based on an application protocol provided by PureTi (Cincinnati, OH, USA). In addition, training was provided by SATA (Spring Valley, MN, USA) for accurate spraying control. The temperature (25 °C) and relative humidity (40–45%) were adjusted to prevent instant evaporation of the sprayed TiO_2_ solution (nanostructured anatase 10 μg/cm^2^ TiO_2_, PureTi, Cincinnati, OH, USA) before application. After cleaning the surface of the panel, the TiO_2_ solution was sprayed. The spray pressure was adjusted to 60 psi with a regulator from the compressor, and the distance between the panel and the spray was ~0.15 m (6 in) at an angle of 90. Coated panels were dried at room temperature for 3 days.

### 2.5. SEM-EDS Analysis of Photocatalyst Coating and Surfaces

The photocatalytic coating was analyzed to analyze the morphology and chemical composition on the surface of common building materials used for livestock barn interiors. Passive treatment of indoor air inside livestock facilities is the ultimate goal for UV treatment. Thus, the surface analyses of how the TiO_2_ coating interacts with common building materials is important, as the mitigation of emissions is, in part, driven by the photocatalyst integrity and uniformity. The scanning electron microscopy-energy-dispersive X-ray spectroscopy (SEM-EDS) analyses were performed at the Materials Analysis and Research Lab, Iowa State University. The SEM-EDS analysis was performed to analyze the TiO_2_ coating morphology on the photocatalyst-coated surfaces. Samples coated TiO_2_ were additionally coated with 2 nm iridium for conductivity and were lightly sprayed with canned air to remove loose dust particles before starting the analysis of SEM-EDS. The samples were examined in an FEI Quanta-FEG 250™ SEM (FEI Company, Hillsboro, OR, USA) 10 kV. A range of magnifications was used. The samples used the electrons (S.E.) imaging and analyzed them in a high vacuum mode for improved resolution. Energy Dispersive Spectroscopy (EDS) analysis was done using an Oxford Instruments Aztec energy-dispersive spectrometer with an X-Max 80 light-element detector (80 mm^2^ active areas) for elemental and chemical analysis of a sample surface. A beam current of ~0.5 nA was used to generate an X-ray count rate of about 15k cps. X-ray maps of 256 × 244 pixels were collected for 10 min to show the distribution of the elements. That also produced a “sum” spectrum showing the overall X-ray signal from the field of view.

In addition, samples used in the previous study [12] were analyzed to compare whether there is a difference according to the material coated with the TiO_2_ photocatalyst. For the TiO_2_ sample coated on the glass used in the previous lab-scale study, samples on the glass were imaged with backscattered electrons (BSE), for which the brightness of the signal correlates with the density/average atomic number of the material for checking the TiO_2_ coating morphology according to the material coated with TiO_2_. In addition, samples were analyzed in variable pressure mode, where 60–100 Pa of water vapor was introduced into the chamber to dissipate the charge. Through this analysis, it was possible to confirm the chemical composition, arrangement, and morphology of the TiO_2_-coated sample surface.

### 2.6. Data Analysis–Effectiveness and Cost of Photocatalytic Treatment of Air

The overall mean percent reduction for each measured gas was estimated as:% R = (C_con_ − C_Treat_)/C_con_ × 100(1)
where: %R = percent reduction in gas concentrations during UV treatment.

C_Con_ & C_Treat_ = the mean measured concentrations in control and treated air, respectively.

Measured gas concentrations were adjusted to standard conditions (defined as 1 atm, 273.15 K) and dry air using collected environmental data:C = C′/(1 − W) × (P∙MW)/(R∙T)(2)
where: C = a standard dry concentration of measured gas (g/m^3^).

C’ = the mean measured gas concentration in control and treated air (mL/m^3^).

W = humidity ratio was calculated with Equation (4) [1,22,23].

MW = molecular weight of target gas (g/mol).

R = 0.082057 L∙atm/(mol∙K).

T = measured air temperature (K).

P = measured pressure (atm).

The measured treated airflow rate was also adjusted to standard dry conditions at both control and treatment sampling locations:Q = Q′ × (1 − W) × (P′ × 273.15 K)/(P × T)(3)
where: Q = dry standard airflow rate (m^3^/min).

Q’ = actual measured (humid) airflow rate (m^3^/min).

W = humidity ratio calculated with Equation (4) [22,23].

P’= actual pressure at the sampling point (atm).

P = standard pressure (atm).

The humidity ratio was estimated as:W = 0.62198 × φ × e^f^(T)/[(Ps × 101325) − φ × e^f^(T)](4)
where: W = humidity ratio (kg of water per kg of dry air).

Ps = pressure at the sampling location (atm).

φ = relative humidity (decimal).

For cases where T < 273.15 K [22,23]:f(T) = c_1_/T + c_2_ + c_3_T + c_4_T^2^ + c_5_T^3^ + c_6_T^4^ + c_7_lnT

For cases where T > 273.16 K [22,23]:f(T) = c_8_/T + c_9_ + c_10_T + c_11_T^2^ + c_12_T^3^ + c_13_lnT
where: c_1_ = −5.565∙10^3^, c_2_ = 6.392, c_3_ = −9.678∙10^−3^, c_4_ = 6.222∙10^−7^, c_5_ = 2.075∙10^−9^, c_6_ = −9.484∙10^−13^, c_7_ = 4.163, c_8_ = −5.800∙10^3^, c_9_ = 1.391, c_10_ = −4.864∙10^−2^, c_11_ = 4.176∙10^−5^, c_12_ = −1.445∙10^−8^, and c_13_ = 6.545.

Gas emissions were calculated as a product of measured gas concentrations and the total airflow rate:E = C × Q(5)
where: E = gas emissions (g/min) of a target pollutant.

C = the mean measured target pollutant gas concentration in control or treated standard dry air (g/m^3^).

Q = dry standard airflow rate (m^3^/min).

The electric energy consumption was calculated using the measured power consumption of UV lamps during treatment. Electric energy consumption (kWh) during treatment was calculated using:EEC = P × t_s_/(3600 × 1000)(6)
where: EEC = electric energy consumption (kWh).

P = measured electric power consumption for the UV lamps turned ‘on’ during treatment (W).

t_s_ = treatment time for air in contact with the UV lamps that were turned ‘on’ inside the mobile lab (s).

The mass of mitigated gas pollutant (M) with UV during given treatment time (ts) was estimated by comparing gas emission rate (E) in treatment and control:M = (E_con_–E_treat_) × t_s_/60(7)
where: M = mass of mitigated gas pollutant (g).

E_con_ = emission rate at the ‘control’ sampling location.

E_treat_ = emission rate at the ‘treatment’ sampling location.

The electric energy of UV treatment (EE, kWh/g) was estimated as using electric energy consumption (EEC) needed to mitigate a gas pollutant mass (M):EE = EEC/M(8) Finally, the estimated cost of electric energy (Cost) needed for UV treatment was estimated using the mean cost ($/kWh) of rural energy in Iowa ($0.13/kWh, [24]):Cost = EE × $0.13/kWh(9)
where: Cost = estimated cost of electric energy needed for UV treatment to mitigate a unit mass of pollutants in the air ($/g).

UV dose was estimated using measured light intensity (I) at a specific UV wavelength (mW/cm^2^) and treatment time (t_s_). Since the photocatalysis reaction is assumed to be the main mechanism for the target gas mitigation, the light intensity irradiated on the TiO_2_ surface was used.
UV dose = I × t_s_(10)
where: UV Dose = energy of the UV light on the surface of photocatalyst (mJ/cm^2^).

### 2.7. Statistical Analysis

The program of R (version 3.6.2, R Studio, Boston, MA, USA) was used to analyze the mitigation of target standard gases under the UV-A photocatalysis treatment. The mitigation depending on parameters of UV dose and treatment time between control concentration and treatment concentration was statistically analyzed using one-way ANOVA. The statistical difference was confirmed by obtaining the p-value through the Tukey test. A significant difference was defined for a *p*-value < 0.05 in this study.

## 3. Results

### 3.1. NH_3_ Percent Reduction in Treated Air–Effect of UV-A Dose Controlled by Treatment Time

The NH_3_ percent reduction (%R) was investigated by increasing the UV dose by controlling the treatment time (Table 2). A 5% NH_3_ standard gas was injected into the filtration unit inlet (Figure 1) and mixed with ambient air resulting in 67.8 ± 0.2 ppm at the inlet to the mobile laboratory. Initial testing used 60 UV lamps installed in 12 chambers (Figure 1); the NH_3_ reduction was investigated by sampling at three different treatment times (from 29 to 57 s). The was no significant reduction in NH_3_ with the largest UV dose tested (2.2 Mj/cm^2^). However, the measured concentrations in the control and treatment were reproducible. This observation led us to explore increasing the UV dose by installing additional UV lamps.

### 3.2. NH_3_ Percent Reduction in Treated Air–Effect of UV-A Dose Controlled by Light Intensity and Time

The NH_3_ percent reduction (%R) was investigated by increasing the UV dose by installing additional lamps (from 60 to a total of 160) and maximizing treatment time (Table 3). The additional LED UV-A lamps (110 lamps) using portable UV lamp holders were installed in two chambers (#2 and #3) (Figure 1), and then the number of lamps turned ‘on’ was controlled.

A statistically significant reduction of 9–11% was measured (Table 3) for UV doses of 3.90 and 5.81 mJ/cm^2^. The extrapolated cost for removing 1 kg of NH_3_ from the air was ~$53–$63. Furthermore, the high light intensity and shorter treatment time were more cost-effective compared with low light intensity and higher residence time.

Measurement of NH_3_ concentration after UV treatment was repeated three times with rapid ‘lamps on’ and ‘lamps off’ showing similar mitigation effects (Figure 5). This finding has practical significance because of the simplicity of activating treatment with no apparent lagtime.

### 3.3. Butan-1-ol Percent Reduction in Treated Air–Effect of UV-A Dose Controlled by Treatment Time

As with NH_3_, there was no significant percent reduction for the initial 60 lamps turned on in 12 chambers (Table 4). A 100 ppm butan-1-ol standard gas was injected into the filtration unit inlet (Figure 1) and mixed with ambient air resulting in 0.63 ± 0.04 ppm at the inlet to the mobile laboratory and similar concentrations after UV treatment. Still, the measured concentrations in the control and treatment were reproducible. This observation led us to explore increasing the UV dose by installing additional UV lamps for this model VOC.

### 3.4. Butan-1-ol Percent Reduction in Treated Air–Effect of UV-A Dose Controlled by Light Intensity and Time

A statistically significant percent reduction (19–41%) in butan-1-ol was found for the UV doses greater than 2.48 mJ/cm^2^ (i.e., when additional lamps were installed, Table 5). The percent reduction for butan-1-ol was higher than for NH_3_. The percent reduction increased with the UV dose, but the 3.90 mJ/cm^2^ appeared to be the most economically efficient (i.e., ~$0.35 to remove/mitigate 1 mg butan-1-ol from the air).

Measurement of butan-1-ol concentration after UV treatment was repeated three times with rapid ‘lamps on’ and ‘lamps off’ showing similar mitigation effects (Figure 6) similar to the effect observed for NH_3_. This finding has practical significance because of the simplicity of activating treatment with no apparent lagtime.

## 4. Discussion

### 4.1. Evaluation of TiO_2_-Based UV-A Photocatalysis

Previous research on the mitigation of selected target gases via photocatalysis with UV-A in livestock-relevant environmental conditions was summarized in Table 6. In the case of NH_3_, the photocatalysis showed a percent reduction from 7% ~ 19% as the light intensity increased in the lab-scale experiment [12]. At the pilot-scale [11], the reduction with photocatalysis efficiency was reduced to ~5% to 9%. Although the detailed mechanism of photocatalysis varies with different target pollutants, it is commonly agreed that the primary reactions responsible are interfacial redox reactions of the electron (e−) and hole (h+) on the surface of the photocatalyst coating material. Therefore, this is considered that inhibiting factors, such as dust and high humidity, can reduce the interfacial redox reactions on the TiO_2_ surface.

In this study, a 9% reduction was observed when the average photocatalysis of light intensity at the photocatalytic surfaces was 0.49 mW/cm^2^. Statistically significant NH_3_ reduction was observed for sufficiently high light intensity even at shorter treatment times (9.5 s). In the environment of livestock facilities, the NH_3_ mitigation using UV-A photocatalysis was found to be less than 20%. This is considered to be less attractive compared to the 50–99% reduction efficiency of other NH_3_ mitigation technologies (dietary additives, manure additive, manure storage handling, and manure incorporation [9,25,26]). Therefore, based on this and previous research, we do not recommend the use of the UV-A photocatalysis technology in the livestock farm for the only purpose of reducing NH_3_.

Depending on the type of VOC, the reduction efficiency varied greatly. It means there was a significant decrease (mitigation) and increase (generation) in some types of VOC. VOCs also showed a higher percent reduction in lab-scale [12,17] experiments compared with the pilot-scale [11]. The photocatalysis showed a percent reduction from 27% ~ 100% in the lab-scale experiment. At the pilot-scale, the reduction with photocatalysis efficiency was reported to be as low as (–53%, generation) to ~62% (mitigation). This decreased percent reduction could result from increased dust and relative humidity for the pilot-scale testing. This study also showed that VOC reduction by UV-A photocatalysis could be reduced with a short treatment time (and therefore the dose), similar to the results of previous studies. The results highlight the requirement to carefully scale up treatments from controlled lab-scale studies into the pilot-scale and eventually on-farm.

### 4.2. Evaluation of TiO_2_ Coated Surfaces with SEM-EDS Analysis

We conducted SEM-EDS analyses to gain insight into the morphology and chemical composition of TiO_2_ photocatalyst and its interaction with common building materials. The results of analyzed TiO_2_ coating morphology on the surface and chemical composition used in this and our previous studies are shown in Table 7. The morphology of coated TiO_2_ was different depending on the surface material. TiO_2_ sprayed on the glass dried in the form of ‘droplets’. The TiO_2_ component was detected only on the white ‘circle’ of dried solid material from evaporated droplets (Table 7 and Figure A8a). Therefore, it is predicted that TiO_2_ and photocatalysis were most active at those selected fractions of the entire surface. However, for the FRP (fiberglass reinforced plastic used for barn construction), the TiO_2_ coating was covering the whole area (Table 7 and Figure A8b). The dose of TiO_2_ coated on the FRP (embossed part) was the same as the TiO_2_ dose coated on the glass (Figure A9). Interestingly, a large amount of TiO_2_ dose was detected in the ‘valley’ formed between the embossed parts (Figure A9c). TiO_2_ in the valley formed a thick ‘cake’, as shown in Figure 7. It is considered likely that the TiO_2_ liquid solution was further ‘drained’ into the valley part of FRP when spraying the solution of TiO_2_. It is recommended to conduct trials of TiO2 application on surfaces to learn the spraying technique and control drying conditions to achieve a practically uniform coating.

Dust accumulation on TiO_2_ coated surfaces can affect treatment effectiveness. Dust and organic substances were detected on the TiO_2_ surface (Table 7). In fact, while most of the dust present on the dust-accumulated TiO_2_ sample was removed with canned air spray before analysis, some dust and organic substances attached to the surface were still detected. The accumulated dust is expected to cover the surface with TiO_2_ (Figure A9b,d). However, the TiO_2_ was also detected on the surface of the sample used in an environment where dust accumulated.

The chemical composition associated with TiO_2_ coating was presented in Figure A9. We observed that the TiO_2_ coated on a glass surface was completely removed with propan-2-ol (isopropyl alcohol, Figure A9e). Therefore, it is considered that care must be taken when cleaning the TiO_2_ coated surface. However, based on the fact that TiO_2_ was detected on the surface after photocatalysis in the TIO_2_ sample used at 60% relative humidity, it is believed that TiO_2_ can operate under high humidity conditions for an extended period of time. However, it is considered that additional experiments are required to test the practical application of TiO_2_ coating inside farms where power-washing with water (and sometimes with disinfectants) is performed periodically or in environments where condensation is formed on the wall and ceiling due to temperature differences inside and outside.

## 5. Conclusions

We designed, tested, and commissioned a mobile laboratory for on-farm research and demonstration of UV treatment for gaseous emissions in real farm conditions. The mobile lab is capable of treating up to 1.2 m^3^/s of air with TiO_2_-based photocatalysis and adjustable UV-A dose based on LED lamps. The commissioning of all systems with standard gases resulted in ~9% and 34% reduction of NH_3_ and butan-1-ol, respectively. We demonstrated that as the percent reduction of standard gases increased with increased UV dose by both increased light intensity and treatment time. The environmental conditions of air flowrate, light intensity, and standard gas blending were reproducible. The estimation of extrapolated costs of mitigating targeted gases was possible. The TiO_2_ coating was able to adhere to common building materials, but the overall coating integrity and practical re-application should be investigated in farm-scale trials. The follow-up trials to verify this technology with the mobile UV laboratory on the farm-scale are warranted.

## Figures and Tables

**Figure 1 ijerph-18-01523-f001:**
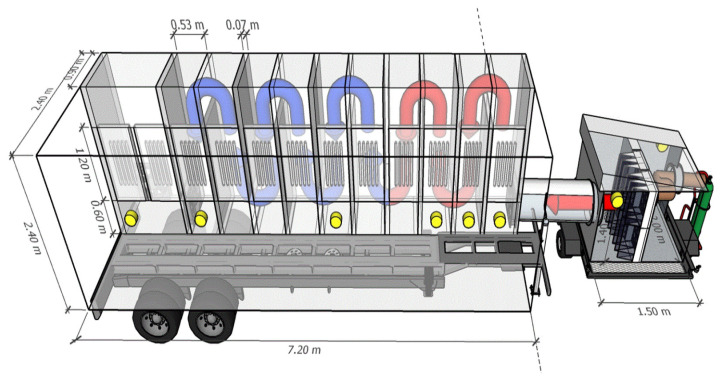
Schematic of a flow-through UV mobile laboratory with an upstream filtration unit. Brown arrow: inlet of untreated air; red arrow: inlet air with reduced particle matter load (after filtration); blue arrow: UV-treated air. The untreated air (brown arrow) could be either standard gas (illustrated by the green compressed gas cylinder), a mixture of standard gases, surrogate odorous air, exhaust from livestock barn, or other air pollution source. Yellow: gas sampling ports used for evaluation of treatment efficiency. Air moves in a serpentine pattern through a series of twelve ‘chambers,’ each equipped with UV sources (shown as the group of five vertical lamps) and clad with surface panels coated with photocatalyst (TiO_2_).

**Figure 2 ijerph-18-01523-f002:**
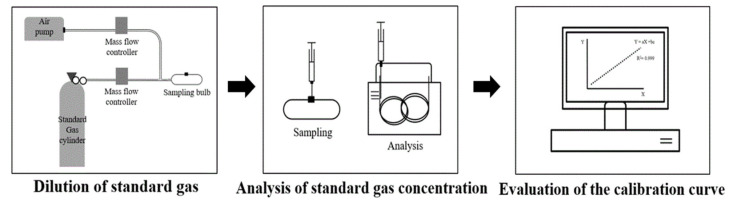
Calibration method for measuring targeted gas concentration. (1) Five standard gas concentrations were prepared/diluted to be within the range of the target gas to be measured. (2) Standard gas samples were analyzed with SPME-GC-MS or electrochemical gas sensors resulting in a gas concentration calibration curve.

**Figure 3 ijerph-18-01523-f003:**
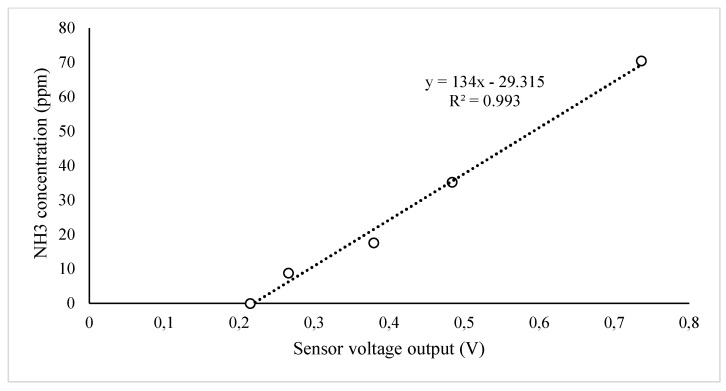
Calibration for real-time NH_3_ measurements.

**Figure 4 ijerph-18-01523-f004:**
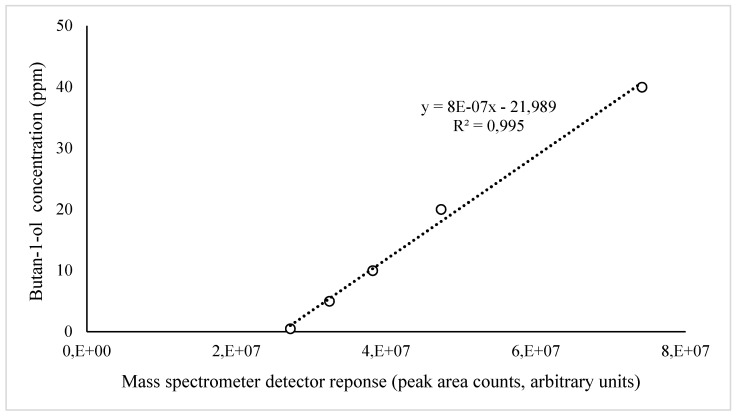
Calibration for butan-1-ol measurements.

**Figure 5 ijerph-18-01523-f005:**
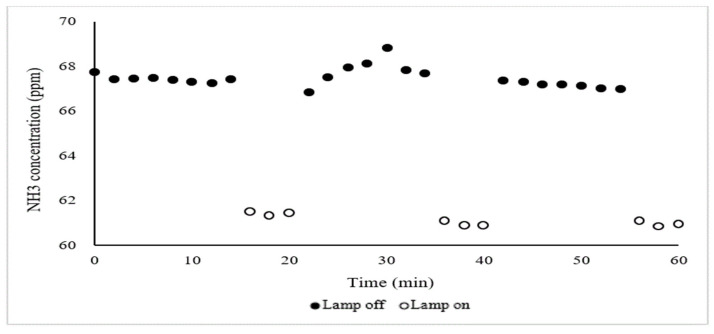
Mitigation of NH_3_ concentration with 110 UV-A lamps inside the two chambers (#2 and #3, Figure 1). NH_3_ concentration was measured at the effluent of chamber #3. Airflow = 0.25 m^3^/s, inlet air temperature (influent of chamber #2) = 13 °C, outlet air temperature = 19 °C, RH = 36% (effluent of chamber #3). The y-axis start at 60 ppm (not 0).

**Figure 6 ijerph-18-01523-f006:**
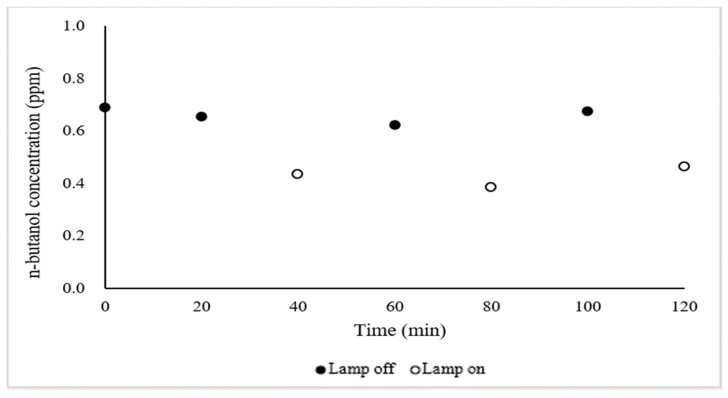
Mitigation of butan-1-ol (a.k.a. n-butanol) concentration with 110 UV-A lamps in the two chambers (#2 and #3). The reduction was measured by adding UV-A lamps inside two chambers. Black means light off, and white means light on. Airflow = 0.25 m^3^/s, inlet air temperature = 13 °C, outlet air temperature = 19 °C, RH = 36% (inside mobile laboratory).

**Figure 7 ijerph-18-01523-f007:**
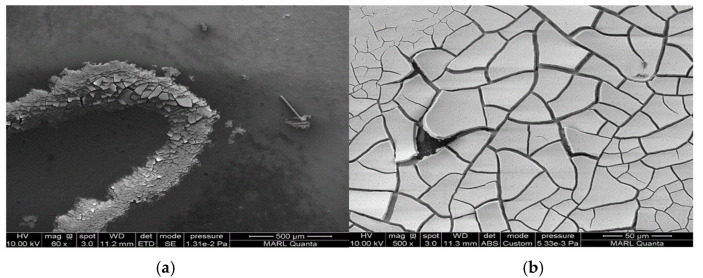
Sprayed TiO_2_ coating arrangement in the ‘valley’ formed between the embossed fiberglass reinforced plastic (FRP). (**a**): 500 μm & (**b**): 50 μm magnification.

**Table 1 ijerph-18-01523-t001:** Requirements for the mobile laboratory to evaluate the effectiveness of UV photocatalysis at a farm-scale. Appendix A.

Requirement	Constraints	Approach	Performance Metric	Detailed Description
(1) Mobile lab for on-site testing of UV treatment on gaseous emissions from livestock barns	Mobility	Repurposed mobile trailer	Can be towed on public roads	Figure 1
				Figure A1
				Appendix A
	The safe work environment for lab personnel to work on-site year-round	(a) Space divided of UV treatment chamber and work area for samples	(a) Can be safely operated (e.g., collecting data) during UV treatment	Figure 1 Figure A6 Figure A7 Appendix A
		(b) Negative-pressure ventilation inside the UV treatment area		
			(b) Maintaining room temperature inside the work area regardless of ambient air	
		(c) Heating, air condition		
		(d) Airtight UV treatment chamber		
		(e) Rodent-proof		
	Installation of coated FRP for photocatalytic reaction	Fixed the coated FRP to all surfaces of the UV chamber with a pushpin	Coated ~76% of the total surface area in each chamber with a photocatalytic coating	Appendix A
	Connectivity to the air pollution source	Large (dia = 0.5 m) flexible ducting for easy connection to barn exhaust fans	Figure A6	Appendix A
			Appendix A	
		Safe routing of the excess of fan exhaust	It cannot affect the barn fan performance	
	‘Plug-and-play’ 110V power management for 50 Amp lab	30 m (grade type) cable with NEMA (type) plug		
	Sufficient, TiO_2_-coated surface for photocatalysis with UV light	Constructed vertical baffles inside the UV treatment chamber	Appendix A	Appendix A
(2) Control the UV dose (via lamps’ power)	Sufficient number of installed lamps to facilitate the photocatalysis reaction	Installed additional UV lamp holders	Can control UV dose (~5.8 mJ/cm^2^, Appendix A	Figure A2
		measuring UV irradiance		Figure A4
				Appendix A
(3) Control the volumetric airflow	Ability to treat ~0.25 to 1.0 m^3^/s of air	(a) Installed two fans and 1 anemometer fan	Can control airflow from ~0.25 m^3^/s (535 CFM) to ~1.23 m^3^/s (2,600 CFM).	Appendix A
		(b) Built a monitor system to see the volumetric flow rate measured by the anemometer fan		
(4) Control the photocatalyst dose	The necessity of coating TiO_2_ on the FPR surface	Coated through precise spray control	Material and Method Section 2.4	Discussion Section 4.2
				Appendix C
(5) Control airborne particulate matter	The necessity to remove airborne substances from the incoming gases for accurate investigation of the reduction effect by photocatalysis	Installed the MERV filtration unit	Appendix A	Figure A5
				Appendix A

FRP = fiberglass reinforced plastic; NEMA = National Electrical Manufacturers Association; CFM = cubic feet per min; MERV = minimum efficiency reporting value.

**Table 2 ijerph-18-01523-t002:** Mitigation of NH_3_ concentration under UV-A photocatalysis with 60 lamps (2.2 mJ/cm^2^). Control (Chamber #1, chamber nearest to the air inlet), C#6, C#10, C#12 (chamber nearest to the air outlet) signifies the location of air sampling ports. Airflow = 0.25 m^3^/s, inlet air temperature = 8 °C, outlet air temperature = 9 °C, RH = 39% (inside mobile laboratory). All (60) LED lamps were ‘on’.

	Control	Chamber Number		
		(Treatment Time, UV Dose)		
		C#6	C#10	C#12
		(29 s, 1.2 mJ/cm^2^)	(48 s, 1.9 mJ/cm^2^)	(57 s, 2.2 mJ/cm^2^)
NH_3_ concentration (ppm)	67.9	67.9	67.4	67.5
	67.6	68.0	67.0	67.0
	67.8	67.6	65.0	63.8
Average ± S.D.	67.8 ± 0.2	67.8 ± 0.2	66.5 ± 1.3	66.1 ± 2.0
(*p*-value)		(0.79)	(0.23)	(0.29)

**Table 3 ijerph-18-01523-t003:** Mitigation of NH_3_ with increasing UV-A light intensity and time. Airflow = 0.25 m^3^/s, temperature= 11 ± 3 °C, RH = 34 ± 6%, number of repeated measurements (n) = 3.

UV Dose, mJ/cm^2^	Measured Gas Concentration		%R ^b^	Pollutant Emission		Power ^c^	Electric Energy for Mitigation of Pollutant Mass ^d^	Cost ^e^
(# Lamps ^a^, Treatment Time, *t_s_*)	(ppm)		(*p*-Value)	(*E*, g/min)		(W)		($/g)
	Control	Treatment		Control	Treatment		(*EE*, kWh/g)	
**0.38**	**67.8**	**67.8**	**0% (0.79)**	0.76	0.76	160	Not estimated	Not estimated
**(10, 9.5 s)**	**±0.17**	**±0.21**						
0.67	67.4	67.4	0% (0.93)	0.74	0.74	470	Not estimated	Not estimated
(40, 9.5 s)	±0.35	±0.42						
1.33 (60, 9.5 s)	67.6 ±0.69	67.4 ±0.35	0% (0.41)	0.74	0.74	790	Not estimated	Not estimated
2.48 (80, 9.5 s)	67.6 ±0.32	66.9 ±0.82	1% (0.36)	0.76	0.74	1260	Not estimated	Not estimated
**3.90** **(110, 9.5 s)**	**67.4** **±0.36**	**61.1** **±0.30**	**9% (<0.01)**	**0.75**	**0.68**	**1730**	**0.41**	**0.05**
**5.81**	**68.9**	**61.1**	**11%**	**0.76**	**0.68**	**2500**	**0.48**	**0.06**
**(160, 57 s)**	**±0.68**	**±0.70**	**(<0.01)**					

^a^ The number of lamps turned ‘on’ during treatment; ^b^ percent reduction in gas concentrations; ^c^ measured electric power consumption for the UV lamps turned ‘on’ during treatment (W); ^d^ The electric energy of UV treatment (EE) estimated as using the electric energy consumption (EEC) needed to mitigate a gas pollutant mass (M) (kWh/g); ^e^ The cost of electric energy needed for UV treatment to mitigate a unit mass of pollutant in the air ($/g); **Bold** font signifies the statistical significance of treatment.

**Table 4 ijerph-18-01523-t004:** Mitigation of butan-1-ol concentration under UV-A photocatalysis with 60 lamps. Control (Chamber #1, chamber nearest to the air inlet), C#6, C#10, C#12 (chamber nearest to the air outlet) signifies the location of air sampling ports. Airflow = 0.25 m^3^/s, inlet air temperature = 11 °C, outlet air temperature = 13 °C, RH = 34% (inside mobile laboratory). All (60) LED UVA lamps were ‘on’.

	Control ppm	Chamber Number		
		(Treatment Time, UV Dose)		
		C#6	C#10	C#12
		(29 s, 1.2 mJ/cm^2^)	(48 s, 1.9 mJ/cm^2^)	(57 s, 2.2 mJ/cm^2^)
butan-1-ol (ppm)	0.59	0.55	0.62	0.63
	0.67	0.66	0.61	0.62
	0.62	0.66	0.63	0.69
Average ± S.D.	0.63 ± 0.04	0.62 ± 0.06	0.62 ± 0.01	0.65 ± 0.04
(*p*-value)		(0.73)	(0.87)	(0.63)

**Table 5 ijerph-18-01523-t005:** Mitigation of butan-1-ol concentration with increasing light intensity. Airflow = 0.25 m^3^/s, temperature = 14 ± 2 °C, RH = 34 ± 6%, number of repeated measurements (n) = 3.

UV Dose mJ/cm^2^	Measured Gas Concentration			Pollutant Emission			Electric Energy for Mitigation of Pollutant Mass ^d^	
(# Lamps ^a^, Treatment Time, *t_s_*)	(ppm)		%R ^b^	(*E*, mg/min)		Power ^c^		Cost ^e^
	Control	Treatment	(*p*-Value)	Control	Treatment	(W)	(*EE*, kWh/mg)	($/mg)
**0.38**	**0.63**	**0.62**	**0% (0.73)**	29.9	29.5	160	Not estimated	Not estimated
**(10, 9.5 s)**	**±0.04**	**±0.63**						
0.67	0.81	0.67	16% (0.33)	38.5	32.1	470	Not estimated	Not estimated
(40, 9.5 s)	±0.27	±0.09						
1.33	0.67	0.60	10% (0.41)	32.1	28.6	790	Not estimated	Not estimated
(60, 9.5 s)	±0.09	±0.03						
**2.48**	**0.66**	**0.53**	**19% (0.04)**	**31.5**	**25.3**	**1260**	**3.40**	**0.44**
**(80, 9.5 s)**	**±0.02**	**±0.06**						
**3.90**	**0.65**	**0.43**	**34% (0.03)**	**30.9**	**20.3**	**1730**	**2.71**	**0.35**
**(110, 9.5 s)**	**±0.03**	**±0.04**						
**5.81**	**0.69**	**0.41**	**41% (0.02)**	**32.9**	**19.4**	**2500**	**3.10**	**0.40**
**(160, 57 s)**	**±0.02**	**±0.07**						

^a^ The number of lamps turned ‘on’ during treatment; ^b^ percent reduction in gas concentrations; ^c^ measured electric power consumption for the UV lamps turned ‘on’ during treatment (W); ^d^ The electric energy of UV treatment (EE) estimated as using the electric energy consumption (EEC) needed to mitigate a gas pollutant mass (M) (kWh/g); ^e^ The cost of electric energy needed for UV treatment to mitigate a unit mass of pollutant in the air ($/mg); **Bold** font signifies the statistical significance of treatment.

**Table 6 ijerph-18-01523-t006:** Summary of percent reduction for NH_3_ and VOCs with TiO_2_ (coating thickness: 10 μg/cm^2^) and UV-A light.

Reference	Experiment Conditions	Treatment Time ^a^ (s)	Light Intensity (mW/cm^2^)	Average Percent Reduction of	
				Targeted Gas	
				NH_3_ (Range)	VOCs (Range)
**[17]**	**Lab-scale**	**40, 200**	**0.06**	**Not reported**	**DMDS (35.0–40.4)**
					**DEDS (27.7–81.0)**
	**Temp ^b^: 40**				**DMTS (37.1–76.3)**
					**BA (62.2–86.9)**
	**RH ^c^: 40%**				**Guaiacol (37.4–100.0)**
					**p-Cresol (27.4–93.8)**
[12]	Lab-scale	40, 200	0.44	**7.3–9.4**	Not reported
	Temp ^b^: 25 ± 3	40, 200	4.85	**10.4–18.7**	Not reported
	RH ^c^: 12%				
[13]	Pilot-scale	24, 47	<0.04	Not reported	**AA** (–**52.9** to –19.7)
	Temp ^b^: 22~26				
	RH ^c^: 36~80%				**p-Cresol** (–21.4–**22.0**)
[11]	Pilot-scale	100, 170	0.44	–0.2–**5.2**	DEDS (12.7–18.7)
					BA (6.1–21.8)
					**p-Cresol** (**32.2**–11.1)
	Temp ^b^: 28 ± 3				Skatole (–35.9–18.5)
	RH ^c^: 56%	40, 170	4.85	2.5–**8.7**	**DEDS** (18.1–**47.2**)
					**BA** (22.1–**61.9**)
					**p-Cresol** (21.8–**49.3**)
					**Skatole** (53.6–**35.4**)
This study	Pilot-scale	9.5	Photolysis ^d^: Ave 0.88	**9.4**	**Butan-1-ol (34.4)**
	Temp ^b^: 19		Photocatalysis ^e^: Ave 0.49		
	RH ^c^: 36%				

^a^ Time to irradiate the target gas with UV-A light; ^b^ Temperature (°C); ^c^ Relative humidity; ^d^ Average of photolysis light intensity measured at three locations (top, middle, bottom); ^e^ Average of photocatalysis light intensity measured at eleven panels; dimethyl disulfide (DMDS), dimethyl disulfide (DEDS), dimethyl trisulfide (DMTS), acetic acid (AA), butanoic acid (BA); **Bold** font signifies a statistical difference in mitigating gases with UV at (*p* < 0.05).

**Table 7 ijerph-18-01523-t007:** TiO_2_ coating morphology on the sprayed surface of common building materials.

Ref.	TiO_2_ Dose	Characteristic	TiO_2_ Arrangement
[12]	10 μg/cm^2^	Coating method: spray TiO_2_ coating surface: glass	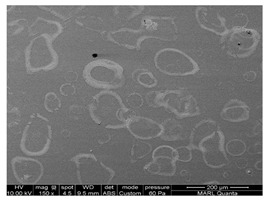
		Coating method: spray TiO_2_ coating surface: glass Poultry dust (black) was accumulated for 1 week	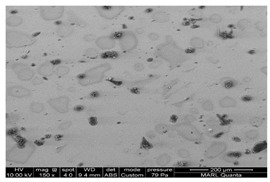
[11]	10 μg/cm^2^	Coating method: spray TiO_2_ coating surface: FRP Poultry dust (black) was accumulated for 2 month	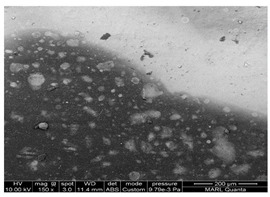
This study	10 μg/cm^2^	Coating method: spray TiO_2_ coating surface: FRP	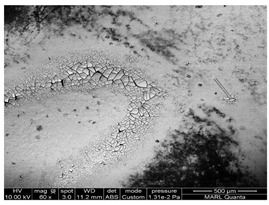

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
