# Peer review of "Design and Testing of Mobile Laboratory for Mitigation of Gaseous Emissions from Livestock Agriculture with Photocatalysis"

_ijerph, 2021, doi:10.3390/ijerph18041523_

Round 1
Reviewer 1 Report
Good research, application, and presentation of an alternative method to treat air emissions from livestock facilities.
In the abstract and within the article, the authors mention that the system was tested under "real farm conditions", such statements can bring false expectations to the reader. Since all the tests were performed using filtered air (not from a livestock facility) and standard gases, that claim is not sustained. If the claim refers to meteorological conditions (done outdoors with similar weather as in a farm at that time), it should specify that. The manuscript describes the calibration and operation readiness to be tried on-farm in future research. This should also be clearly stated in Materials and Methods point 2.3 and conclusions.
Some acronyms need to be defined at first use.
In figure 1, points b, c, and d are not identified in the graphic
On Materials and Methods point 2.3 add verbiage referring the reader to the Supplemental Material for details on fans installation and operations. As described now, it is not clear how the air is moving (and airflow controlled) within the unit unless you read the supplemental material at that point.
Check for the correct use of subscripts on chemical and mathematical formulas within the text and tables.
Were all variations of light intensity and retention time repeated three times (3 repeated measures)? This is evident in tables 2 and 4 and figures 5 and 6 and their explanation. The number of repeated measures (n) needs to be mentioned in the verbiage related to tables 3 and 5. At a minimum, 3 repeated measures are needed.
The same situation with point 3.4 on the Butan-1-ol runs description.
The reduction in testing gases' concentration under some of the tested parameters is statistically significant in the trial. This allows for certification of the operative conditions of the unit. Net results are on the lower end of reduction compared to other on-farm technologies. Reduction of air exhaust from livestock facilities is the next step, as well as expanding the "odorous compounds" measures.
On Suplemental Material
Line 74 (FigureA.21) doesn't exist. Should it be Figure A3?
Reviewer 2 Report
There have been laboratory scale studies of photocatalysis to treat emissions but fewer larger scale tests. The authors have constructed a mobile laboratory to examine the photocatalytic treatment of emissions from livestock production under real farm conditions. They used UV-A (365 nm) which is safer than lower wavelength UV. They describe their mobile laboratory and its performance in reducing ammonia and butan-1-ol through UV treatment.
This research study appears to have been competently performed. It is very interesting and it will be very useful to those working to develop photocatalysis emission control systems for farming and other applications. It is well written and I support its publication. I suggest one minor modification.
Suggestion: Line 168: Please subscript the 2 in TiO2.
Reviewer 3 Report
General:
Thank you for this interesting research about the possibility for the mitigation of gaseous emissions
from livestock agriculture, which is a big environmental problem.
Major revisions:
1) You cite mainly literature from your own group. Is this really the only group working in the
field? If not, please cite also results from other groups and compare your results to their
results in the discussion section.
2) Ammonia and other greenhouse emission from livestock agriculture are a big environmental
problem and different possibilities for mitigation are under discussion. Please discuss the
percentage of reduction that is possible with your technology in comparison with other
techniques
(Lit for ammonia e.g. Bittman, S., Dedina, M., Howard C.M., Oenema, O., Sutton, M.A., (eds),
2014, Options for Ammonia Mitigation: Guidance from the UNECE Task Force on Reactive
Nitrogen, Centre for Ecology and Hydrology, Edinburgh, UK
Chaopu Ti, Longlong Xia, Scott X. Chang, Xiaoyuan Yan (2019): Potential for mitigating global
agricultural ammonia emission: A meta-analysis, Environmental Pollution, Volume 245, Pages
141-148)
3) In the abstract, you write, ‘In this paper, we present the design, testing, and commissioning
of a mobile laboratory for on-farm research and demonstration of performance in real farm
conditions.’ I therefore expected that you show results from the test of the mobile
laboratory as well with standard gas as well as with air from real farm buildings. In the text
however, you show only results from experiments with standard air. Please change the
sentence in the abstract.
Additionally it is not clear to me, why you need a MERV filter setup for the experiments
described in the paper, because you only describe experiments with standard air. Please
explain.
4) It is not clear to me, what is the final plan for using your method of mitigation in real farm
conditions. Would it be to put a TiO2 coating to the surfaces inside the barns (because you
tested the coating of common building material) or do you want to use the method outside
the farm buildings with an equipment like your MERV filter and the lab wit h12 chambers?
Please explain why you tested the TiO2 coating of common building material.
5) In Figure 1, you show gas-sampling ports at the lower end of chambers #1, #3, #6, #10 and
#12. This corresponds with the results in Table 2 and 4. Under Figure 5, you write that you
sampled the influent of chamber #2 and at the effluent of chamber #3. How is this possible?
6) Additionally you write under Figure 5 and 6 that the temperature was 13 °C at the influent of
chamber #2 and 19 °C at the effluent of chamber #3. What caused this increase of the
temperature? Is this just due to the higher temperature in you mobile Lab or is this due to
the UV-light? If so, what would this mean for using the method inside CAFOs?
Minor revisions:
- Material and methods: Please explain why you choose a concentration of 67. 8 ppm for
ammonia and 0.63 ppm for butan-1-ol as control concentration in your experiments. Are
they typical concentrations for CAFOs? For which animals? Are they near limit values?
- Table 1, line 114: Please explain also the abbreviations NEMA and CFM = cubic feet per
minute
- Figure 2: Please consider deleting. It is not necessary in my opinion.
- Page 7, equation (3): Please insert K after 273.15
- Page 7, line 218-229: Please check units in equation (4) and in line 224-229
- Page 9, line 273-274: Please consider deleting these two sentences. I think they are not
necessary
- Figure 5: Please make it explicit, that the y-axis does not start at 0. Perhaps with a
interrupted y-axis.
- Page 11, line 338: Please change to … was higher than for NH3.
- Page 12, line 355: replace “measurement” by “measured”
- Page 13, Table 6: In your other studies [10, 11, 12, 16] you used other VOCs. Why did you
change to Butan-1-ol in this study? Please explain in the materials and methods chapter.
- Page 13, Table 6: add “°C” after the temperatures in column 2
- Page 13, line 386: Could the decreased percent reduction be caused by the lower
temperature in this study?
